# Preoperative Classification of Peripheral Nerve Sheath Tumors on MRI Using Radiomics

**DOI:** 10.3390/cancers16112039

**Published:** 2024-05-28

**Authors:** Christianne Y. M. N. Jansma, Xinyi Wan, Ibtissam Acem, Douwe J. Spaanderman, Jacob J. Visser, David Hanff, Walter Taal, Cornelis Verhoef, Stefan Klein, Enrico Martin, Martijn P. A. Starmans

**Affiliations:** 1Department of Surgical Oncology and Gastrointestinal Surgery, Erasmus MC Cancer Institute University Hospital Rotterdam, Dr. Molewaterplein 40, 3015 GD Rotterdam, The Netherlands; i.acem@erasmusmc.nl (I.A.); c.verhoef@erasmusmc.nl (C.V.); 2Department of Plastic and Reconstructive Surgery, University Medical Center Utrecht, Heidelberglaan 100, 3584 CX Utrecht, The Netherlands; e.martin-2@umcutrecht.nl; 3Department of Radiology & Nuclear Medicine, Erasmus MC Cancer Institute University Hospital Rotterdam, 3015 GD Rotterdam, The Netherlands; x.wan@erasmusmc.nl (X.W.); d.spaanderman@erasmusmc.nl (D.J.S.); j.j.visser@erasmusmc.nl (J.J.V.); d.hanff@erasmusmc.nl (D.H.); s.klein@erasmusmc.nl (S.K.); m.starmans@erasmusmc.nl (M.P.A.S.); 4Department of Neurology, Erasmus MC Cancer Institute University Hospital Rotterdam, Dr. Molewaterplein 40, 3015 GD Rotterdam, The Netherlands; w.taal@erasmusmc.nl; 5Department of Pathology, Erasmus MC Cancer Institute University Hospital Rotterdam, 3015 GD Rotterdam, The Netherlands

**Keywords:** machine learning, magnetic resonance imaging, radiomics, radiologic imaging, soft tissue sarcoma

## Abstract

**Simple Summary:**

This study aims to improve the preoperative classification of nerve sheath tumors using radiomics, a method that extracts quantitative data from medical images. By analyzing MRI scans, we seek to develop a more accurate way to distinguish between different types of nerve sheath tumors before surgery. Our findings could lead to better treatment planning and outcomes for patients with these tumors. This research has the potential to enhance the diagnostic process and contribute to more personalized care for individuals with nerve sheath tumors, ultimately benefiting the medical community and patients alike.

**Abstract:**

Malignant peripheral nerve sheath tumors (MPNSTs) are aggressive soft-tissue tumors prevalent in neurofibromatosis type 1 (NF1) patients, posing a significant risk of metastasis and recurrence. Current magnetic resonance imaging (MRI) imaging lacks decisiveness in distinguishing benign peripheral nerve sheath tumors (BPNSTs) and MPNSTs, necessitating invasive biopsies. This study aims to develop a radiomics model using quantitative imaging features and machine learning to distinguish MPNSTs from BPNSTs. Clinical data and MRIs from MPNST and BPNST patients (2000–2019) were collected at a tertiary sarcoma referral center. Lesions were manually and semi-automatically segmented on MRI scans, and radiomics features were extracted using the Workflow for Optimal Radiomics Classification (WORC) algorithm, employing automated machine learning. The evaluation was conducted using a 100× random-split cross-validation. A total of 35 MPNSTs and 74 BPNSTs were included. The T1-weighted (T1w) MRI radiomics model outperformed others with an area under the curve (AUC) of 0.71. The incorporation of additional MRI scans did not enhance performance. Combining T1w MRI with clinical features achieved an AUC of 0.74. Experienced radiologists achieved AUCs of 0.75 and 0.66, respectively. Radiomics based on T1w MRI scans and clinical features show some ability to distinguish MPNSTs from BPNSTs, potentially aiding in the management of these tumors.

## 1. Introduction

Peripheral nerve sheath tumors (PNSTs) are soft-tissue tumors that are associated with a number of peripheral nerves, encompassing a variety of both benign and malignant tumors [1]. Among benign peripheral nerve sheath tumors (BPNSTs), schwannomas are the most prevalent type (up to 80%), with neurofibromas constituting a significant portion of the remaining BPNSTs (10–24%) [2,3]. Malignant peripheral nerve sheath tumors (MPNSTs) are rare, accounting for 2% of all soft-tissue sarcoma (STS) [4]. About 23–51% of MPNSTs occur in neurofibromatosis type 1 (NF1) patients. These tumors can be sporadic or radiation-induced as well [5,6,7,8]. NF1 patients have an 8–13% lifetime risk of developing an MPNST, which is the leading cause of mortality among these patients [9,10]. High-grade MPNSTs behave aggressively, with a high risk of metastasis and recurrence. Around 60% of patients develop metastases over time and approximately 40–70% of patients experience recurrence [5,6,7,11,12,13,14,15,16,17,18]. The prognosis of MPNSTs remains poor, as the median 5-year survival rates of localized disease at diagnosis ranges between 23–69% [19].

Therapeutic strategies of PNSTs vary depending on tumor type and grade. The treatment for asymptomatic BPNST consists of monitoring; however, in symptomatic cases, the tumor can be removed through intracapsular resections, minimizing neurological damage [20,21]. In contrast, the resection of high-grade MPNSTs commonly results in significant postoperative morbidity and major nerve deficits [22]. Distinguishing MPNSTs from BPNSTs is crucial for appropriate treatment planning, so biopsies are essential for all instances of suspected lesions. However, biopsy, although standard practice, is invasive, challenging, and associated with pain and nerve damage. Moreover, sampling errors may occur due to the heterogeneous nature of PNSTs, such as neurofibromas, when they undergo dedifferentiation [23].

Magnetic resonance imaging (MRI) plays a central role in the evaluation and monitoring of PNSTs, offering detailed imaging that aids in assessment. Despite its importance, there are currently no known distinctive MRI features that allow radiologists to establish a certain diagnosis between MPNSTs and BPNSTs [2]. Even for experienced radiologists, the accuracy in distinguishing between these tumor types may only be around 50% [24,25]. Ultrasound (US) is often the initial imaging modality for STS due to its availability, portability, and cost-effectiveness, allowing real-time image acquisition to confirm lesions, differentiate solid from cystic masses, and assess blood flow. However, conventional US has limited accuracy in distinguishing between benign and malignant STTs, leading to the use of advanced techniques like elastography and contrast-enhanced ultrasound (CEUS) to improve diagnostic capabilities [26]. In addition to MRI, diffusion-weighted imaging (DWI) and positron emission tomography (PET) have been reported as useful supplementary imaging modalities, providing additional information on tumor cellularity and metabolism. However, potential risks and imprecision are associated with these modalities [2,27,28,29]. Thus, a novel method to improve the discrimination between MPNSTs and BPNSTs is needed.

Radiomics is a quantitative approach to identifying significant imaging features from medical images, which has been widely used in current tumor research [24,30,31,32,33,34]. These quantitative features have the potential to uncover tumoral patterns and characteristics that are subtle for visual inspection and to avoid subjective assessment by radiologists. Integrated with machine learning techniques, radiomics can be used to find optimized combinations of imaging features and construct diagnostic models. Studies have shown the potential of combining radiomics and machine learning in differentiating between MPNSTs and BPNSTs using MRI [34,35]. These approaches show promise in reducing unnecessary biopsies and alleviating anxiety. However, existing studies primarily focus on a single MRI scan, leaving the potential of combining multiple MRIs unexplored. Additionally, region of interest (ROI) segmentation, a labor-intensive task in radiomics research, often requires time-consuming manual efforts. While manual segmentation ensures accuracy, it may not always be feasible within radiologists’ workflow. The potential of automatic or semi-automatic segmentation tools needs further investigation.

Therefore, this study aims to develop a radiomics model to differentiate MPNSTs from BPNSTs, evaluating various combinations of MRI scans and clinical features. The performance of the radiomics model will be compared with radiologists in this task and the study will also assess the efficacy of manual versus semi-automatic segmentation methods.

## 2. Materials and Methods

### 2.1. Study Population and Data Collection

Patients referred to or treated at Erasmus Medical Center (Rotterdam, The Netherlands) between 2000 and 2019, with a confirmed pathological diagnosis of either MPNST or BPNST (neurofibroma or schwannoma), and who had either undergone surgical treatment or had at least 1 year of follow-up without malignant transformation, were included if they had non-metastatic disease at the time of diagnosis and had undergone a MRI scan as part of standard care. The MRI scans included T1-weighted (T1w) imaging, T2-w imaging, T1 with fat saturation and gadolinium (T1w-FS-GD), T1 Spectral Presaturation with Inversion Recovery with gadolinium (T1w-SPIR-GD), T2 with fat saturation (T2w-FS), and T2 with Short τ Inversion Recovery (T2w-STIR). Notably, several MRI scans were conducted in hospitals that referred these patients, resulting in differences in scanner types and scan protocols. No restrictions on the acquisition protocol were imposed, which could result in not all sequences being present for each patient. All the MRI scans were pseudonymized. This study received approval and review from the local medical ethics committee (MEC-2019-0843), with a waiver of informed consent. 

For each identified patient, clinical features were extracted from medical records, including age at operation, sex, neurogenetic diagnosis (i.e., NF1 or Schwannomatosis), presence of spontaneous pain, and presence of preoperative motor deficits.

### 2.2. Manual and Semi-Automatic Segmentation

Each lesion was manually segmented once by one of two observers (I.A. and E.M.) on the MRI scan where the tumor was deemed best visible to indicate the ROI. These segmentations were randomly distributed between the observers. Random samples were verified with a musculoskeletal radiologist. To transfer the segmentation from one image sequence to the others, all image sequences were spatially aligned with the sequence where the initial segmentation was performed. This alignment was achieved through automated image registration using a rigid transformation model and mutual information metric with the Elastix software (https://sourceforge.net/projects/elastix/) [36]. Following image registration, a visual inspection was carried out on all segmentations to verify the quality of the aligned images. Images on which segmentation after registration showed zero overlap with the tumor, due to misregistration, were excluded.

As manual segmentation is labor-intensive and time-consuming in clinical practice, we also evaluated a semi-automated minimally interactive segmentation method called InteractiveNet [37]. InteractiveNet requires the user to only input the six extreme points per tumor (i.e., two in each direction), which are used to guide a convolutional neural network to perform the segmentation. This framework’s efficacy has been previously validated across thirteen types of soft tissue tumors but we are the first to test its efficacy in MPNSTs and neurofibromas.

In the context of this study, each lesion was semi-automatically segmented on T1w MRI by one observer (I.A.) using InteractiveNet. Subsequently, the observer scored the resulting segmentation quality based on four scales (Excellent, Sufficient, Insufficient, and Incorrect, see Appendix A) [37]. Similar to the manual segmentations, the semi-automatic segmentations were spatially aligned to the other sequences using image registration.

### 2.3. Radiomics Feature Extraction

For each lesion on each MRI scan, a total of 564 radiomics features were extracted. An overview of all features is provided in Appendix A. Radiomics features quantifying intensity, shape, and texture were extracted by the Workflow for Optimal Radiomics Classification (WORC) toolbox (version 3.6.3), which internally uses the PREDICT (version 3.1.17) and PyRadiomics software (version 3.0.1) [38,39,40,41]. More details on the extracted features can be found in the WORC paper [41]. The categorical clinical features were converted into numerical form.

### 2.4. Decision Model Creation

The WORC algorithm was used to find the optimal decision model from the extracted features. An overview of the methodology implemented in WORC is presented in Figure 1. Within WORC, the creation of decision models unfolds in multiple stages, e.g., feature imputation, feature scaling, feature selection, and feature resampling and classification.

Throughout each step, a variety of algorithms and their associated hyperparameters are considered and an exhaustive search is conducted to determine the combinations of algorithms that yield the highest prediction accuracy on the training set. Recognizing that a single optimal solution might be a coincidental finding, the 100 highest-performing decision models are combined into a single model by ensembling.

### 2.5. Classification by Radiologists

To compare radiomics with clinical practice, two musculoskeletal radiologists (8 and 9 years of experience in differentiating BPNSTs from MPNSTs), specialized in the evaluation of STS, scored the lesions on a four-point scale to indicate their level of confidence regarding whether the tumor was an MPNST (ranging from 1, indicating strong disagreement, to 4, indicating strong agreement). Lesions with scores above 2 were classified as MPNST, while those with scores equal to or below 2 were considered BPNST. The radiologists were blinded for the diagnosis. To assess the impact of clinical features on the diagnosis, each radiologist scored lesions twice: once without and once with the clinical features.

### 2.6. Experimental Set-Up

First, to evaluate the predictive capacity of different combinations of MRI scans for MPNST, multiple radiomics models were developed. Three models were constructed exclusively using the non-contrast enhanced T1w MRI and T2w MRI scans, either as separate or combined inputs. Next, several models based on combinations of these with the other MRI scans (e.g., T1w MRI with contrast) were constructed and evaluated. In the cases where a specific MRI scan is missing for a tumor, automatic imputation of feature values was employed within the WORC pipeline to compensate for the missing sequence.Second, for the T1w MRI models, the above experiment was repeated using radiomics features extracted from the semi-automatic segmentations to verify whether a more time-efficient approach could replace manual segmentation while maintaining prediction accuracy. To evaluate the impact of segmentation quality, the performance of the model trained and tested on all semi-automatic segmentations was compared to the model both trained and evaluated only using the tumors with semi-automatic segmentations scored as “Sufficient” or “Excellent” in quality. 

Third, to examine the added value of clinical features on top of imaging, a model that combined clinical features with the radiomics model with the highest mean AUC was evaluated. Additionally, two models were developed, with one utilizing only clinical features and the other using only volume, to evaluate the individual predictive value of these features.

Fourth, the predictions from the best radiomics model and radiologists were merged. This integration was realized by using either the OR or AND operator between the binarized predictions from the model and those made by the radiologist. In the OR integrated model, a tumor is classified as malignant if either the model or the radiologist indicates malignancy. Conversely, in the AND integrated model, a tumor is classified as malignant only if both the model and the radiologist indicate malignancy. The radiomics model with the best AUC was integrated with two radiologists in these two different ways, resulting in a total of four models created during this step.

### 2.7. Statistical Analysis and Evaluation

All statistical analyses on patient demographics were performed in R (version 4.2.2). Baseline characteristics were compared between patients with MPNSTs or BPNSTs. The chi-squared test and Fisher exact test were used to analyze categorical variables. The Student’s *t*-test was used to analyze continuous variables. All analyses with a *p* < 0.05 were considered statistically significant. The performance of models was assessed through random-split cross-validation, as is the default in WORC [41]. The dataset was randomly divided into 80 percent for training and 20 percent for testing and 100 iterations were conducted to construct confidence intervals on the performance estimates (Appendix A). Within each iteration, the training set underwent an internal five-times random-split cross-validation. All model optimization of WORC was performed exclusively within the training set, followed by evaluation on the testing set. 

All statistical analyses on the imaging characteristics and radiomics models were performed in Python (version 3.6.8). The area under the curve (AUC) of the receiver operating characteristic (ROC) curve, accuracy, sensitivity, and specificity were used for the evaluation. These metrics were computed by averaging over the 100 cross-validation iterations. Furthermore, 95% confidence intervals (95% CI) for the performance measures were computed using the corrected resampled *t*-test [42]. To construct 95% CIs for the performance of radiologists, 1000× bootstrap resampling was employed [43]. Cohen’s kappa was used to measure the agreement between radiologists.

To gain insight into which radiomics features are most informative for prediction, univariate statistical testing using the Mann–Whitney U test was applied on the radiomics features. Bonferroni correction was used to correct *p*-values of features from multiple testing, i.e., multiplying the *p*-values by the number of tests.

## 3. Results

### 3.1. Clinical Characteristics of Database

A total of 35 (32%) MPNSTs and 74 (68%) BPNSTs were included. Patient characteristics and clinical features are summarized in Table 1. In both patient groups with MPNSTs and BPNSTs, there were fewer male patients, comprising 30% and 43%, respectively. Additionally, in both MPNST and BPNST groups, most patients experienced spontaneous pain (70% and 59%, respectively). Furthermore, the proportion of patients who experienced pre-operative motor deficits in MPNSTs was significantly higher than that in BPNSTs (33% versus 12%, *p* = 0.02). MPNSTs were generally larger than BPNSTs, with a mean volume of 208 cubic centimeters (cm^3^*)* compared to 54 cm^3^ (*p* = 0.01).

An overview of the included MRI scans and the variation in the acquisition protocols is depicted in Appendix A. The dataset originated from 37 different scanners, including 23 model types (Siemens: 27 patients, Philips: 32 patients, and General Electric: 50 patients), and thus showed substantial heterogeneity in the acquisition protocols.

### 3.2. Evaluation of the Radiomics Models

A comprehensive overview of the performance of all radiomics models is shown in Table 2. Among the models with manual segmentations, their performance was similar, with mean values of metrics close to each other and confidence intervals substantially overlapping. In particular, the T1w MRI model exhibited the highest mean AUC of 0.71.

Among the models with semi-automatic segmentations, the T1w MRI model had slightly lower AUC compared to the T1w MRI model with manual segmentations (0.68 versus 0.71). Notably, using only the segmentations with “Sufficient” or “Excellent” quality scores resulted in similar performance of the radiomics model.

The T1w MRI model with manual segmentations, as the radiomics model with the highest mean AUC, was selected for integration with clinical features. The performance of the T1w MRI model combined with clinical features, the imaging-only T1w MRI model, the clinical model, and the volume model are presented in Table 3, with the corresponding ROC curves of the first two models depicted in Figure 2. The integrated T1w MRI model with clinical features exhibited a slightly higher AUC than the imaging-only T1w MRI model (0.74 versus 0.71). The model relying only on volume showed a higher mean AUC than the model relying solely on clinical features (0.64 versus 0.60) but substantially lower than the radiomics model.

Within the radiomics model of the highest mean AUC, using T1w MRI and clinical features, only two T1w imaging features showed statistically significant predictive power (Appendix A). Specifically, these features are the major axis length and the maximum 2D diameter (column). The major axis length represents the largest axis length of the ROI-enclosing ellipsoid and the maximum 2D diameter denotes the largest pairwise distance between tumor surface mesh vertices in the column-slice plane [40].

### 3.3. Evaluation of the Radiologists

The performance of the radiologists is presented in Table 3 and Figure 2. When assessing the tumor without the clinical features, Radiologist 1 had a better performance than Radiologist 2 (AUC: 0.79 versus 0.68). Confidence intervals of the AUCs from both radiologists overlap substantially. In the second assessment with clinical features, both radiologists’ mean AUCs slightly decreased (0.75 and 0.66, respectively). Interestingly, adjustments made by both radiologists in the second assessment resulted in more incorrect predictions, mainly involving the changes from benign to malignant classifications (Appendix A). Cohen’s kappa values between the two radiologists were computed as 0.46 with clinical features and 0.47 without, suggesting a moderate level of interobserver agreement.

In both assessments, the mean AUC values of Radiologist 1 were higher than those of T1w MRI models, while the mean AUC values of Radiologist 2 were lower than T1w MRI models.

### 3.4. Evaluation of the Integrated Models

An overview of the performance of integrated models is shown in Table 4. Both integrated models using the OR operation showed a higher mean sensitivity compared to the performance of radiologists. However, the mean accuracy and mean specificity of the integrated models were lower than those of the radiologists before integration. Conversely, both integrated models using the AND operation showed higher mean accuracy and mean specificity than the radiologists before integration. Due to the nature of the AND operation, which requires stricter conditions for predicting malignancy compared with the OR operation, the mean sensitivity decreased in the two models using this operation.

## 4. Discussion

In this study, radiomics models were developed and evaluated to differentiate between BPNSTs and MPNSTs using preoperative MRI scans and clinical features. The best-performing radiomics model, combining both T1w MRI scans and clinical features, was, to some degree, able to distinguish MPNSTs from BPNSTs. Adding other MRI scans did not yield an improvement in performance. Moreover, the model’s performance was further compared with two radiologists, where one performed slightly better than the model, while the other performed slightly worse. Lastly, the T1w MRI radiomics model using semi-automatic segmentation achieved a slightly lower AUC compared to manual segmentation but significantly reduced manual workload.

Our study investigated radiomics models utilizing various combinations of MRI scans and clinical features for the diagnosis of MPNST. First, T1w and T2w MRI models demonstrated comparable diagnostic power. However, combining T1w and T2w MRI scans did not enhance model performance in our results, which indicated that similar diagnostic information was used from both scans. This may be explained by the fact that only shape features were found to be significant in univariate testing in both T1w and T2w MRI. Notably, two shape features (MajorAxisLength and Maximum2DDiameterColumn) associated with tumor diameters emerged as significant predictors, aligning with findings from previous studies [34,44]. Second, incorporating clinical features in the radiomics model slightly improved the mean AUC, while the model using only clinical features exhibited some predictive power but it was not substantial. Using clinical features thus might add some value to the radiomics models.

Furthermore, our study revealed the varying interpretations of BPNSTs and MPNSTs among different radiologists, as well as inconsistent evaluations of the same tumor between two assessments. Analyzing the altered evaluations that led to new errors in the second round, we observed that errors mainly come from benign tumors with above-average volume and malignant tumors with below-average volume. These findings underscore the challenges of achieving consistent and convincing agreement on the diagnostic outcomes of PNSTs, even among experienced radiologists. Additionally, using clinical information might have limited ability to mitigate these challenges in diagnosing complex tumors.

In comparison to radiologists, all the presented radiomics models exhibited higher specificity but much lower sensitivity. This discrepancy could potentially result in missed diagnoses of MPNSTs, leading to delays in crucial treatments in clinical practice. Achieving a high sensitivity is crucial for diagnosing MPNSTs, which indicates that the current models might not be sufficient for clinical support. However, adjusting the choice of the point along the ROC curve allows for fine-tuning the balance between sensitivity and specificity, providing flexibility in model performance tailored to specific clinical needs. Furthermore, radiomic models offer the advantage of providing relatively consistent predictions regardless of experience or training for years. Hence, our study attempted to integrate the opinions of radiologists and the radiomics models using two logical operations. Although neither method improved the overall performance, integrating the assessments from radiomics models and radiologists could be a potential approach to address this challenge for future work.

Prior studies have underscored that radiomics is a promising non-invasive method for distinguishing between BPNSTs from MPNSTs [34,35]. Zhang et al. identified 95 MPNSTs and 171 BPNSTs using T1-GD scans, achieving an AUC of 0.85 with a model incorporating both clinical and imaging features [34]. Similarly, Ristow et al. developed a radiomics model in 17 MPNSTs, 117 BPNSTs, and 8 atypical neurofibromas among NF1 patients, achieving an AUC of 0.94 using fat-suppressed T2-weighted MRI scans without clinical features [35]. Our models did not achieve comparable performances. One potential reason for this discrepancy could be differences in data sources and acquisition protocols. Zhang et al. [34] used data from three institutes, while Ristow et al. [35] collected data from a single institute with a standardized acquisition protocol. In contrast, our dataset exhibited heterogeneity due to differences in acquisition protocols, as part of the data was obtained from other institutes. Second, the limited availability of MRI scans could have affected the model’s performance. Ristow et al. [35] achieved a model with a higher AUC using fat-suppressed T2w MRI compared to Zhang et al. [34] using T1-GD. However, our study did not observe such a performance difference. Since T1-GD or fat-suppressed T2w was not available for all patients, T1-GD and fat-suppressed T2w scans were therefore compared indirectly when combined with both T1w and T2w scans. Although data imputation on radiomics features was employed to compensate for missing scans, the limited availability of MRI scans may have affected the comparison between models.

With respect to the existing studies, our study brings several novel contributions to this field. First, instead of focusing on a single MRI scan, we explored the potential of radiomics models using different combinations of MRI scans through comprehensive performance comparisons. Second, a semi-automatic segmentation method was introduced, which could provide a notable reduction in time compared to manual segmentation, particularly for tumors in less visible positions. Although the T1w MRI model with semi-automatic segmentations achieved through this method exhibited slightly lower AUC than the T1w MRI model with manual segmentation, the efficiency gained in terms of time could make this approach valuable, especially for future pilot studies. Third, the model creation process in our study involves a comprehensive and automated search for optimized models and the corresponding parameters, different from studies using a single-model approach with manual optimization. Last, our cross-validation scheme completely eliminates the risk of overfitting, which is a common concern in radiomics models.

Several limitations in this study should be noted. First, the variation in the imaging protocols might have influenced the performance of the models. Our study did not impose any restrictions on the parameters of MRI acquisition settings, as selecting a single protocol is unrealistic in clinical practice and could have compromised the robustness of radiomic models. Second, our dataset is a retrospective collection, which might introduce potential bias. This bias might particularly occur in the BPNSTs group, as tumors reaching a certain volume or presenting symptoms like pain might be more likely to be referred to a tertiary center. Moreover, this bias may also add difficulties to the classification task in our dataset, as these referred cases may be more complex. Lastly, missing data were present in our study as there are no standardized imaging modalities for the diagnosis of MPNSTs across every hospital. While data imputation on the radiomics features was employed to compensate for the absence of actual data, these absent data may have (negatively) affected the performance.

In future research, there are several avenues worth exploring further. First, investigating radiomics in other imaging modalities may provide valuable insights. FDG-PET serves as an additional imaging modality for distinguishing between neurofibromas and MPNSTs in NF1 but not in sporadic cases [28,45,46,47]. Multiple attempts have been made to identify optimal semi-quantitative parameters for distinguishing malignant transformation, e.g., Standardized Uptake Value (SUVmax); however, this parameter remains a challenge due to the variability in cut-off values [48]. Furthermore, DWI has been proposed as a supplementary sequence offering discriminative information, with BPNSTs typically exhibiting a higher apparent diffusion coefficient than MPNSTs [45,49,50]. Second, leveraging deep learning models may hold promise for enhancing diagnostic capabilities. However, the rarity of PNSTs may pose challenges in obtaining sufficiently large datasets to train deep learning models. One potential direction could involve tuning existing deep learning models trained on MRI scans for other tasks to address our research question. Lastly, the validation of developed models in real clinical settings by the end-users, including clinicians and radiologists, is crucial to evaluate their impact on diagnostic and therapeutic decision making.

## 5. Conclusions

This study investigated the diagnostic potential of radiomics models for distinguishing between benign and malignant PNSTs using multiple MRI scans and clinical parameters. The radiomics model using T1w MRI scans and clinical features showed some ability to distinguish MPNSTs from BPNSTs, with the shape features emerging as predictive, and achieved a comparable AUC to the experienced radiologist. Future refinements, including integration with radiologist assessments, exploration of additional imaging modalities, and leveraging deep learning techniques, hold promise for improving diagnostic accuracy in PNSTs.

## Figures and Tables

**Figure 1 cancers-16-02039-f001:**
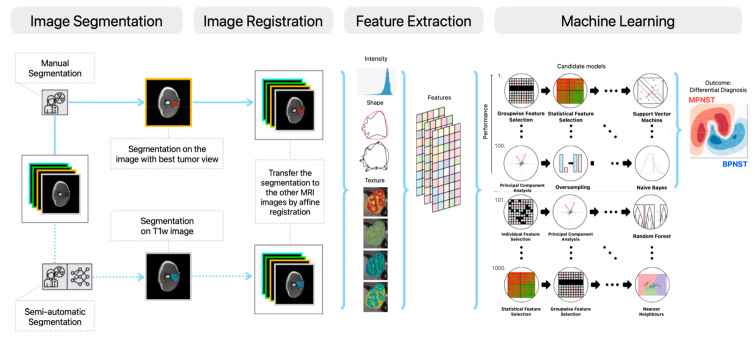
Schematic overview of the radiomics approach, adopted from Vos et al. (2019) [24]. The MRI data is the input data. The processing steps include segmentations of tumors either by manual segmentation or semi-automatic segmentation, image registration, feature extraction, and the creation of machine learning models, consisting of the 100 best workflows from 1000 candidate workflows.

**Figure 2 cancers-16-02039-f002:**
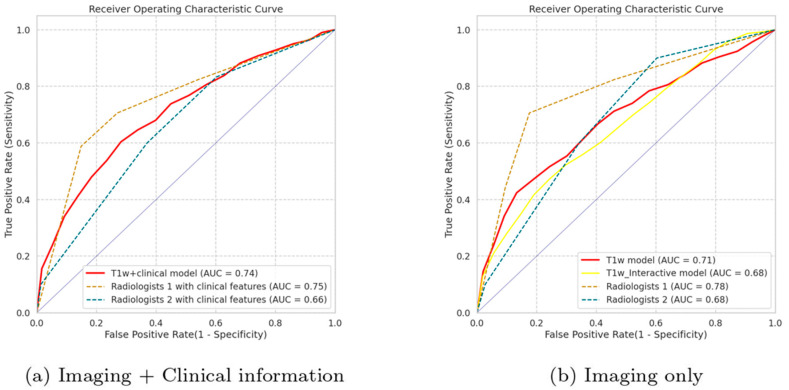
Plots of receiver operating characteristic curves of the radiomics models using T1w MRI and the radiologists employing all MRI sequences (**a**) with using clinical features and (**b**) without using clinical features. T1w + clinical model: the radiomics model with T1w imaging, manual segmentations, and clinical features; T1w model: the radiomics model with T1w imaging and manually created segmentations; T1w_Interactive model: the radiomics model with T1w images and semi-automatic segmentations made using InteractiveNet. Abbreviation: AUC, area under curve.

**Table 1 cancers-16-02039-t001:** Patient characteristics and clinical features of 74 BPNSTs and 35 MPNSTs.

Variable	Benign (N = 74)	Malignant (N = 35)	*p*-Value
Mean Age (SD)	44 yr (19 yr)	43 yr (25 yr)	0.78
Gender Male Female	22 (30%) 52 (70%)	15 (43%) 20 (57%)	0.26
Neurogenetic diagnosis None NF1 Schwannomatosis	35 (47%) 38 (52%) 1 (1%)	16 (47%) 18 (53%) 0 (0%)	0.79
Spontaneous pain No Yes Unknown	28 (41%) 40 (59%) 6	10 (30%) 23 (70%) 2	0.40
Pre-operative motor deficits No Yes Unknown	60 (88%) 8 (12%) 6	22 (67%) 11 (33%) 2	0.02
Mean Volume (SD)	54 cm^3^ (141 cm^3^)	208 cm^3^ (324 cm^3^)	0.01

Unknown denotes missing clinical information. Abbreviations: BPNST, Benign Peripheral Nerve Sheath Tumors; MPNST, Malignant Peripheral Nerve Sheath Tumors; SD, standard deviation; NF, neurofibromatosis; N, number; cm^3^, cubic centimeters.

**Table 2 cancers-16-02039-t002:** The performance of all radiomics models is based on different combinations of MRI sequences.

Models	AUC	Accuracy	Sensitivity	Specificity
**Models with manual segmentations**				
T1w (*n* = 90)	0.71 [0.56, 0.86]	0.75 [0.67, 0.84]	0.30 [0.10, 0.50]	0.93 [0.84, 1.00]
T2w (*n* = 87)	0.70 [0.56, 0.84]	0.65 [0.56, 0.75]	0.33 [0.14, 0.52]	0.83 [0.71, 0.95]
T1w, T2w (*n* = 97)	0.68 [0.54, 0.83]	0.72 [0.64, 0.79]	0.29 [0.11, 0.46]	0.91 [0.82, 0.99]
T1w, T2w, T1w-FS-GD *OR* T1w-SPIR-GD (*n* = 99)	0.70 [0.57, 0.83]	0.72 [0.66, 0.79]	0.26 [0.09, 0.42]	0.92 [0.85, 1.00]
T1w, T2w, T2w-FS *OR* T2w-STIR (*n* = 103)	0.66 [0.55, 0.77]	0.67 [0.60, 0.74]	0.25 [0.08, 0.42]	0.88 [0.78, 0.98]
T1w, T1w-FS-GD *OR* T1w-SPIR-GD (*n* = 93)	0.70 [0.56, 0.83]	0.73 [0.66, 0.80]	0.26 [0.11, 0.41]	0.95 [0.87, 1.00]
T2w, T2w-FS *OR* T2w-STIR (*n* = 94)	0.66 [0.56, 0.77]	0.68 [0.60, 0.75]	0.26 [0.10, 0.42]	0.87 [0.76, 0.97]
**Models with semi-automatic segmentations using InteractiveNet**				
T1w_Interactive model (*n* = 87)	0.68 [0.56, 0.79]	0.70 [0.63, 0.77]	0.23 [0.07, 0.38]	0.94 [0.85, 1.00]
T1w_Interactive_Sufficient model (*n* = 66)	0.64 [0.48, 0.79]	0.69 [0.59, 0.79]	0.20 [0.01, 0.38]	0.89 [0.76, 1.00]

Values are mean [95% confidence interval] over the cross-validation iterations for radiomics models. T1w_Interactive model: the T1w radiomics model using semi-automatic segmentations; T1w_Interactive_Sufficient model: the T1w radiomics model using only semi-automatic segmentations with “Sufficient” or “Excellent” quality scores. Abbreviation: FS, Fat Saturation; *n*, number; SPIR, Spectral Presaturation with Inversion Recovery; GD, Gadolinium Contrast; STIR, Short τ Inversion RecoveryAUC, area under curve.

**Table 3 cancers-16-02039-t003:** Performance of the radiomics models based on T1w MRI and the performance of the radiologists.

Models	AUC	Accuracy	Sensitivity	Specificity
**Imaging + Clinical features**				
T1w + clinical model (*n* = 90)	0.74 [0.60, 0.88]	0.75 [0.69, 0.82]	0.31 [0.12, 0.50]	0.92 [0.84, 1.00]
Radiologist 1 with clinical features (*n* = 108)	0.75 [0.65, 0.85]	0.72 [0.64, 0.80]	0.71 [0.56, 0.86]	0.73 [0.63, 0.83]
Radiologist 2 with clinical features (*n* = 108)	0.66 [0.55, 0.77]	0.62 [0.53, 0.71]	0.60 [0.43, 0.77]	0.63 [0.52, 0.74]
**Imaging only**				
T1w model (*n* = 90)	0.71 [0.56, 0.86]	0.75 [0.67, 0.84]	0.30 [0.10, 0.50]	0.93 [0.84, 1.00]
Radiologist 1 (*n* = 108)	0.78 [0.68, 0.88]	0.79 [0.71, 0.87]	0.71 [0.56, 0.86]	0.82 [0.73, 0.91]
Radiologist 2 (*n* = 108)	0.68 [0.58, 0.78]	0.64 [0.55, 0.73]	0.60 [0.42, 0.78]	0.66 [0.55, 0.77]
**Clinical features only**				
Clinical model (*n* = 90)	0.60 [0.46, 0.73]	0.71 [0.64, 0.78]	0.26 [0.07, 0.44]	0.88 [0.80, 0.96]
**Volume only**				
Volume model (*n* = 90)	0.64 [0.50, 0.78]	0.71 [0.63, 0.80]	0.28 [0.07, 0.49]	0.88 [0.75, 1.00]

Values are mean [95% confidence interval] over the cross-validation iterations for radiomics models and the point estimation on the performance with confidence interval over bootstrap resampling iterations for radiologists. T1w + clinical model: the radiomics model with T1w MRI, manual segmentations, and clinical features; T1w model: the radiomics model with T1w MRI and manual segmentations. Abbreviation: AUC, area under curve.

**Table 4 cancers-16-02039-t004:** Performance of the integrated models based on OR and AND operations.

Metrics	AUC	Accuracy	Sensitivity	Specificity
**OR operation**				
T1w + clinical model *OR* Radiologists 1 (*n* = 90)	N/A	0.68 [0.65, 0.70]	0.74 [0.70, 0.78]	0.65 [0.63, 0.68]
T1w + clinical model *OR* Radiologists 2 (*n* = 90)	N/A	0.63 [0.61, 0.65]	0.71 [0.67, 0.75]	0.61 [0.58, 0.63]
**AND operation**				
T1w + clinical model *AND* Radiologist 1 (*n* = 90)	N/A	0.78 [0.76, 0.79]	0.21 [0.18, 0.25]	0.96 [0.96, 0.97]
T1w + clinical model *AND* Radiologist 2 (*n* = 90)	N/A	0.75 [0.74, 0.76]	0.20 [0.17, 0.24]	0.93 [0.91, 0.94]

Values are mean [95% confidence interval] over bootstrap resampling iterations for integrated models. Specifically: T1w + clinical model *OR* Radiologist 1: Integration of outcomes from the radiomics model utilizing T1w MRI and clinical features with those from Radiologist 1 the OR operation. T1w + clinical model *OR* Radiologist 2: Integration of outcomes from the radiomics model utilizing T1w MRI and clinical features with those from Radiologist 2 using the OR operation. T1w + clinical model *AND* Radiologist 1: Integration of outcomes from the radiomics model utilizing T1w MRI and clinical features with those from Radiologist 1 using the AND operation. T1w + clinical model *AND* Radiologist 2: Integration of outcomes from the radiomics model utilizing T1w MRI and clinical features with those from Radiologist 2 using the AND operation.

## Data Availability

The data that support the findings of this study are available upon reasonable request from the corresponding author. The data are not publicly available due to information that could compromise the privacy of research participants.

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
