# Peer review of "Preoperative Classification of Peripheral Nerve Sheath Tumors on MRI Using Radiomics"

_cancers, 2024, doi:10.3390/cancers16112039_

Round 1

Reviewer 1 Report

Comments and Suggestions for Authors

The authors presented an extremely clear approach to the topic (different types of segmentation—manual and semi-automatic— and the characteristics of the decision model creation process features radiomic's based) that helps to understand the future use of these imaging interpretation tools. A well-conducted retrospective study that describes the lights and proposes the limits of the achieved results.

Author Response

Reviewer #1
The authors presented an extremely clear approach to the topic (different types of segmentation—manual and semi-automatic— and the characteristics of the decision model creation process features radiomic's based) that helps to understand the future use of these imaging interpretation tools. A well-conducted retrospective study that describes the lights and proposes the limits of the achieved results.

  • We would like to thank the reviewer once again for the careful reading of the manuscript. As there are no suggestions of comments, we did not change the manuscript.

Reviewer 2 Report

Comments and Suggestions for Authors

The authors present a study on the use of radiomics in differentiating between benign and malignant peripheral nerve tumors. The topic of the study is very interesting and has a significant impact on the clinical applications for preoperative diagnosis of these tumors. Peripheral nerve tumors, although rare, can be difficult to manage without proper imaging studies, particularly in the case of malignancy. Based on the findings of the study, the authors suggest a combined use of imaging features and clinical symptoms for achieving a higher specificity on MRI evaluation. Although sensitivity rates still need to be improved and MRI aspects may still cause confusion and contradictory opinions on interpretation even for experienced radiologists, the study provides valuable insight into the MRI evaluation of these tumors.

The paper is well-written, easy to follow and comprehensive. Some minor suggestions:

-in the methods section it would be useful to mention whether the radiologists specialized in the evaluation of soft tissue tumors (many countries or centers have radiologists that do not specialize on specific regions or patgologies, making this paper particularly interesting for these professionals)

— in table 1, although it is obvious that there were both female and male patients, it would help the reader to have both genders listed.

Author Response

Reviewer #2
The authors present a study on the use of radiomics in differentiating between benign and malignant peripheral nerve tumors. The topic of the study is very interesting and has a significant impact on the clinical applications for preoperative diagnosis of these tumors. Peripheral nerve tumors, although rare, can be difficult to manage without proper imaging studies, particularly in the case of malignancy. Based on the findings of the study, the authors suggest a combined use of imaging features and clinical symptoms for achieving a higher specificity on MRI evaluation. Although sensitivity rates still need to be improved and MRI aspects may still cause confusion and contradictory opinions on interpretation even for experienced radiologists, the study provides valuable insight into the MRI evaluation of these tumors.

  • We would like to thank the reviewer once again for the careful reading of the manuscript.

The paper is well-written, easy to follow and comprehensive. Some minor suggestions:

  1. In the methods section it would be useful to mention whether the radiologists specialized in the evaluation of soft tissue tumors (many countries or centers have radiologists that do not specialize on specific regions or pathologies, making this paper particularly interesting for these professionals).
  • We have revised this sentence in the Materials and Methods (p.4, lines 177-181): To compare radiomics with clinical practice, two musculoskeletal radiologists (8 and 9 years of experience in differentiating BPNSTs from MPNSTs), specialized in the evaluation of STS, scored the lesions on a four-point scale to indicate their level of confidence regarding whether the tumor was an MPNST (ranging from 1 indicating strong disagreement, to 4 indicating strong agreement).
  1. In table 1, although it is obvious that there were both female and male patients, it would help the reader to have both genders listed.
  • We have added the female gender to the existing table 1 (p. 6, line 253):

Gender

Male

Female

22 (30%)

52 (70%)

15 (43%)

20 (57%)

Reviewer 3 Report

Comments and Suggestions for Authors

This is an interesting manuscript on radiomics and neural tumors. Interestingly Authors conclude that it is too early to affirm it is 100% safe to use this method instead of traditional methods (radiologist and biopsy).

I would add in the introduction a sentence with the role of ultrasound and US with contrast medium in the diagnosis of these tumors.

I would specify if the MRI machine was always the same in all the patients and its specific characteristics (model, Tesla, producer,...)

Author Response

Reviewer #3

This is an interesting manuscript on radiomics and neural tumors. Interestingly Authors conclude that it is too early to affirm it is 100% safe to use this method instead of traditional methods (radiologist and biopsy).

  • We would like to thank the reviewer once again for the careful reading of the manuscript.
  1. I would add in the introduction a sentence with the role of ultrasound and US with contrast medium in the diagnosis of these tumors.
  • We have added the following sentence to the Introduction (p.2, lines 78-83): ‘Ultrasound (US) is often the initial imaging modality for STS due to its availability, portability, and cost-effectiveness, allowing real-time image acquisition to confirm lesions, differentiate solid from cystic masses, and assess blood flow. However, conventional US has limited accuracy in distinguishing between benign and malignant STTs, leading to the use of advanced techniques like elastography and contrast-enhanced ultrasound (CEUS) to improve diagnostic capabilities.

  1. I would specify if the MRI machine was always the same in all the patients and its specific characteristics (model, Tesla, producer,...)
  • The data is a retrospective collection from a tertiary center. The patients may have been initially diagnosed at local medical centers and then referred to our center. As a result, the modalities used on each patient are not streamlined, and there are no restrictions on the parameters of MRI acquisition settings. Besides the already provided information on the scan protocols, we have added the magnetic field strength to our summary of included sequences and their available key parameters in the supplements (Table S2). Additionally, we have mentioned the scanner manufacturers and number of models in the main text (p.6, lines 247-249): The dataset originated from 37 different scanners, including 23 model types (Siemens: 27 patients, Philips: 32 patients, General Electric: 50 patients), and thus showed substantial heterogeneity in the acquisition protocols.’

Additional Note:

Finally, we have critically reviewed the references and made necessary adjustments. Consequently, we revised a sentence in the introduction (p.2, lines 60-62): "The prognosis of MPNSTs remains poor, as the median 5-year survival rates of localized disease at diagnosis range between 23% and 69%."
